# A biomimetic neural encoder for spiking neural network

Shiva Subbulakshmi Radhakrishnan [1], Amritanand Sebastian [1], Aaryan Oberoi [1], Sarbashis Das [2] & Saptarshi Das [1,3,4 ✉]

Spiking neural networks (SNNs) promise to bridge the gap between artificial neural networks (ANNs) and biological neural networks (BNNs) by exploiting biologically plausible neurons that offer faster inference, lower energy expenditure, and event-driven information processing capabilities. However, implementation of SNNs in future neuromorphic hardware requires hardware encoders analogous to the sensory neurons, which convert external/internal stimulus into spike trains based on specific neural algorithm along with inherent stochasticity. Unfortunately, conventional solid-state transducers are inadequate for this purpose necessitating the development of neural encoders to serve the growing need of neuromorphic computing. Here, we demonstrate a biomimetic device based on a dual gated MoS$_2$ field effect transistor (FET) capable of encoding analog signals into stochastic spike trains following various neural encoding algorithms such as rate-based encoding, spike timing-based encoding, and spike count-based encoding. Two important aspects of neural encoding, namely, dynamic range and encoding precision are also captured in our demonstration. Furthermore, the encoding energy was found to be as frugal as ≈1–5 pJ/spike. Finally, we show fast (≈200 timesteps) encoding of the MNIST data set using our biomimetic device followed by more than 91% accurate inference using a trained SNN.

[1] Department of Engineering Science and Mechanics, Pennsylvania State University, University Park, PA, USA. [2] Department of Electrical Engineering, Pennsylvania State University, University Park, PA, USA. [3] Department of Materials Science and Engineering, Pennsylvania State University, University Park, PA, USA. [4] Materials Research Institute, Pennsylvania State University, University Park, PA, USA. ✉email: sud70@psu.edu

Billions of neurons coupled through trillions of synapses form the complex computational unit of the brain, that can simultaneously process a massive amount of information received from external/internal stimuli through various sensory organs. These neurons use action potential or spikes as the common language to speak with each other and to compute for learning and decision-making. Spikes are stereotypical electrical impulses or all-or-none (digital) point events in time, that allow long-distance neural communication and energy-efficient neural computation. However, external stimuli such as light, sound, smell, temperature, etc., and internal stimuli such as blood pressure, oxygen levels, feeling of pain and hunger, etc. that are primarily analog continuous variables in time. It requires specialized sensory neurons, also known as afferent neurons, to transform the specific type of analog stimulus into corresponding spike trains following one or more neural encoding algorithms, and subsequently relay the spike-encoded information to the central nervous system for processing. This process is referred to as sensory transduction. For example, in the auditory neural pathways, mechanoelectrical transduction is mediated by the hair cells within the ear[1]. Similarly, gustatory receptors in taste buds interact with chemicals in food to produce action potentials[2,3]. Phototransduction in the retina is mediated by rods and cones and eventually converted to spikes by the ganglion cells[4,5]. Sensory transduction also exhibits inherent stochasticity, which allows neurons to process information with better noise tolerance and energy efficiency[6,7].

The diversity of neurobiological architectures and neural computational algorithms found inside even the simplest of animal brains continue to fascinate computer scientists and electronic device engineers. Neuromorphic computing pioneered by Carver Mead and colleagues is a branch of research that aims to mimic the computational power of the brain on a chip[8,9]. Unfortunately, the initial growth in neuromorphic computing was rather slow owing to the contemporary dominance of von Neuman architecture, and the success of the complementary metal-oxide-semiconductor (CMOS) technology. However, the recent demise in scaling and fundamental limitations of von Neuman computing is fueling the resurgence of bio-inspired neuromorphic hardware[10–12]. Artificial neural networks (ANNs) are the most prevalent form of neuromorphic computing that have already demonstrated breakthrough progress in many fields[13]. ANNs consist of multiple layers with each layer comprising of collection of computational units, called artificial neurons, which are connected through artificial synapses. While the models of cortical hierarchies from biological neural networks (BNNs) have been mimicked through deep learning[14] in ANNs with a massive number of computational layers, only marginal similarities with brain-like computing can be recognized at the implementation level. The most obvious difference is that artificial neurons receive, process, and transmit analog information in continuous time, whereas biological neurons use action potential or spikes. Also, stochasticity is an inherent neural phenomenon, which is typically ignored by most ANNs. Spiking neural network (SNN) promises to bridge this gap by adopting a new computing paradigm based on biologically plausible neurons[15,16]. In fact, the past few years have seen tremendous progress in the development of SNNs offering unprecedented energy efficiency and faster inference owing to event-driven computation[17]. However, hardware realization of SNNs necessitates the development of neural encoders since conventional sensors are incapable of converting sensory input into spike trains.

Here, we report a biomimetic device based on a dual gated MoS$_2$ field effect transistor (FET) with a stochastic sampling terminal capable of encoding analog signals, for example illuminance levels of a light emitting diode (LED), into corresponding spike trains. We are also able to implement various neural encoding algorithms, such as rate-based encoding, spike timing-based encoding, and spike count-based encoding. Two key features of neural encoding, namely, dynamic range and encoding precision are also captured in our demonstration. Finally, we show a fast and accurate inference of spike encoded MNIST data set using a trained spiking neural network (SNN) with inference accuracy of more than 91%. Remarkably, energy consumption by our biomimetic neural encoder was found to be as frugal as ≈1–5 pJ/spike.

## Results

The overall philosophy of biomimetic neuromorphic computing is shown in Fig. 1. Figure 1a shows the schematic of a biological neural network that involves sensory transduction of analog stimulus to corresponding spike trains by specialized neurons and subsequent processing by the central nervous system. For example, external optical stimuli are converted into corresponding graded potentials by the photoreceptor cells (rods and cones) in the human eyes followed by neural encoding into spike trains using the ganglion cells, and eventual processing of the encoded visual stimulus by the visual cortex. Figure 1b shows the corresponding neuromorphic hardware comprising of neuromorphic sensors, neuromorphic encoders, and neuromorphic processors. Figure 1c shows the schematic of our experimental demonstration with a white light-emitting diode (LED) as the visual stimulus, a silicon (Si) photodiode (PD) as the sensor, a dual gated MoS$_2$-based FET as the neuromorphic encoder, and a trained spiking neural network (SNN) as the neuromorphic processor.

**Biomimetic neural encoder and neuromorphic transducer.** We have used multilayer exfoliated MoS$_2$ that belongs to the family of two-dimensional (2D) layered materials[18–20] as the semiconducting channel, 285 nm SiO$_2$ on p$^{++}$-Si as the back-gate stack, and 120 nm of hydrogen silsesquioxane (HSQ) as the top-gate dielectric for the fabrication of the biomimetic neural encoder as shown schematically in Fig. 2a. The thickness of the MoS$_2$ flake is ≈5 nm. The optical image of the device is shown in Fig. 2b. Source, drain, and gate metal stacks were patterned using electron-beam lithography followed by the deposition of 40 nm of Nickel (Ni) and 30 nm of gold (Au) using electron-beam evaporation. More details on device fabrication can be found in the "Methods" section and in our prior work[21,22]. The channel length and width were ≈1 μm and ≈2.8 μm, respectively. Note that the use of ultrathin body MoS$_2$ as the semiconducting channel material is motivated by the growing interest in 2D materials, as a successor to Si as well as their promising use in neuromorphic and biomimetic devices[12,22–25]. Furthermore, various types of sensors such as photodetectors[26], chemical sensors[27], biological sensors[27], touch sensors[28], and radiation sensors[29] have been demonstrated using MoS$_2$ based devices, which allow direct integration of sensors and encoders in future neuromorphic hardware. The presynaptic signal obtained from the neuromorphic sensors such as the Si PD is applied to the back-gate terminal as analog voltage ($V_{PSV}$), whereas, encoded information in form of postsynaptic current spikes ($I_{PSC}$) is obtained at the drain terminal. The top-gate voltage ($V_{TG}$) is applied as a sequence of sampling pulses, with a pulse duration ($t_p$) of 10 ms and amplitude determined based on the desired encoding algorithm.

Figure 2c shows the transfer function of the neural encoder i.e., $I_{PSC}$ vs. $V_{PSV}$ measured at a drain bias, $V_{DS} = 1$ V, for different $V_{TG}$. The n-type unipolar characteristics is common for MoS$_2$ FETs[30,31]. The monotonic positive shift in the transfer function with decreasing $V_{TG}$ can be explained form the principle of charge balance, i.e., the inversion charge induced by positive $V_{PSV}$

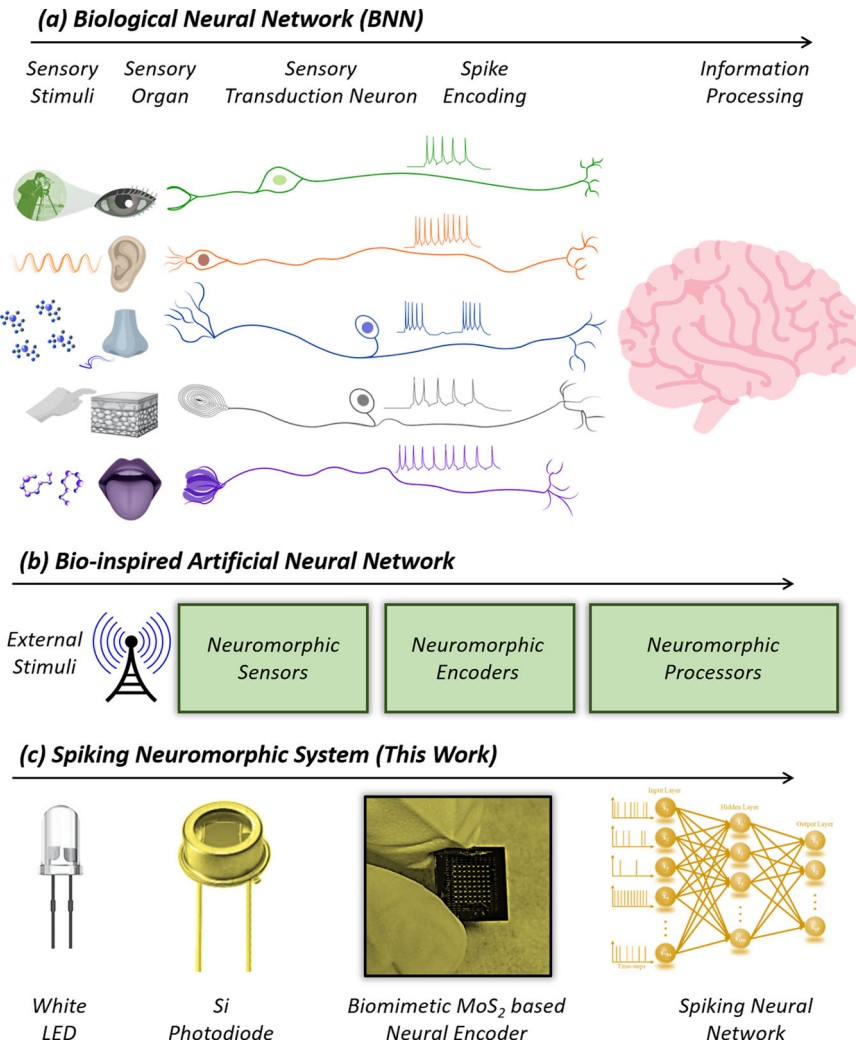

**Fig. 1 Biomimetic neuromorphic computing. a** Schematic of biological neural network (BNN). Specialized neurons convert analog external/internal stimuli into corresponding spike trains by a process called sensory transduction and subsequent relay the information to the central nervous system for further processing. Various encoding algorithms are found in sensory neurobiology such as spike rate-based, spike count-based, and spike timing-based encoding. **b** Schematic of bio-inspired artificial neural network comprising of neuromorphic sensors, neuromorphic encoders, and neuromorphic processors. **c** Schematic of our spiking neuromorphic system consisting of a white light emitting diode (LED) as the visual stimulus, a silicon (Si) photodiode (PD) as the sensor, a dual gated $MoS_2$ based field effect transistors (FETs) with a stochastic terminal as the neuromorphic encoder, and a trained spiking neural network (SNN) as the neuromorphic processor.

is compensated by the negative $V_{TG}$ and vice versa. Figure 2d shows the spiking threshold ($V_{ST}$), which we define as $V_{PSV}$ required to invoke a current spike, i.e., $I_{PSC} > I_{ST}$ as a function of $V_{TG}$, and for different thresholding current, $I_{ST}$. As expected, the spiking threshold is higher for more negative $V_{TG}$ and higher $I_{ST}$. Note that the slope of $V_{ST}$ vs. $V_{TG}$ is constant, irrespective of $I_{ST}$ and is proportional to the ratio of back-gate capacitance to top-gate capacitance, i.e., $C_{TG}/C_{BG}$ which was found to be ≈2.2, consistent with the thicknesses of ≈120 nm and ≈285 nm and dielectric constants of ≈3.2 and ≈3.9 of HSQ and $SiO_2$, respectively. Also, note that the presynaptic terminal (back-gate) and encoding terminal (top-gate) can be interchanged (see Supplementary Fig. 1 and Supplementary Note 1). Figure 2e shows the circuit schematic for phototransduction comprising of a Si PD and a load resistor ($R_L$). Figure 2f shows the current ($I_{PD}$) vs. voltage ($V_{PD}$) characteristics of the neuromorphic sensor, i.e. Si PD for different intensities of the visual stimulus, i.e., LED illuminance ($P_{LED}$). Finally, Fig. 2g shows the phototransduction characteristics of the Si PD and corresponding $V_{PSV}$ applied to the neuromorphic encoder as a function of $P_{LED}$ obtained using $R_L$.

**Hardware acceleration of various neural encoding algorithms.** Next, we implement various neural encoding algorithms, found in sensory neurobiology, using our biomimetic encoder for translating analog $V_{PSV}$ values obtained for different LED illuminations shown in Fig. 3a into corresponding spike trains. The most popular encoding principle is rate-based encoding, originally demonstrated by Adrian and Zotterman[32] using an electrophysiological experiment in sensory nerve fibers of frog muscles. In rate encoding, it is postulated that the information about the stimulus is contained in the firing rate of the neuron, and not in individual spikes. This is more so because the sequence of spikes generated by the neurons in response to a given stimulus varies from trial to trial and over time owing to the inherent stochasticity in sensory transduction, whereas the mean firing rate, i.e., inverse of interspike interval remains practically constant. Numerous studies in sensory and motor systems of various species have validated the spike rate encoding hypothesis. Based on these observations, rate encoding is widely used for SNNs. For rate-based encoding using our biomimetic encoder, the magnitude of $V_{TG}$ pulses are randomly sampled from a Gaussian

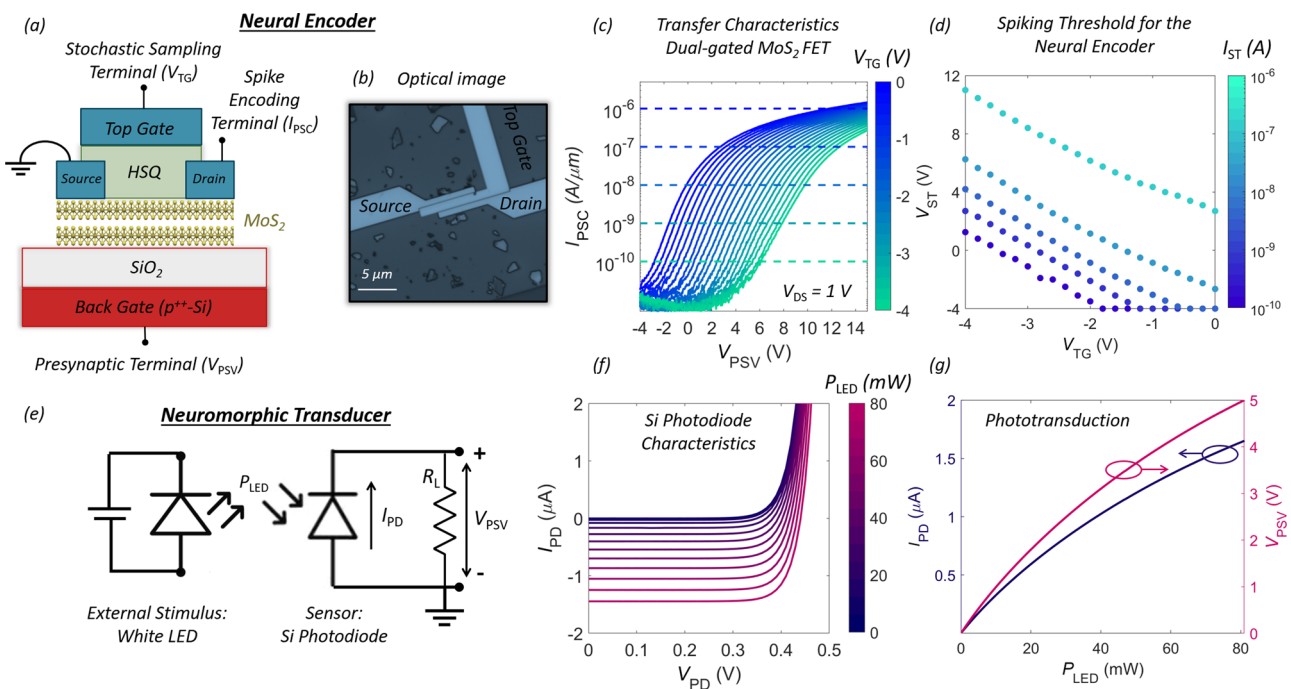

**Fig. 2 Biomimetic neural encoder and neuromorphic transducer. a** Schematic and **b** optical image of our biomimetic neural encoder. We have used multilayer exfoliated $MoS_2$ as the semiconducting channel, 285 nm $SiO_2$ on $p^{++}$-Si as the back-gate stack, and 120 nm of hydrogen silsesquioxane (HSQ) as the top-gate dielectric for the fabrication of the encoder. The channel length and width are $\approx 1$ μm and $\approx 2.8$ μm, respectively. Presynaptic signal obtained from the neuromorphic sensors is applied to the back-gate terminal as analog voltage ($V_{PSV}$), whereas encoded information in form of postsynaptic current spikes ($I_{PSC}$) is obtained at the drain terminal. The top-gate voltage ($V_{TG}$) is applied as a sequence of sampling pulses, with a pulse duration of $t_p = 10$ ms and amplitude determined based on the desired encoding algorithm. **c** Transfer function of the neural encoder i.e., $I_{PSC}$ vs. $V_{PSV}$ measured at a drain bias, $V_{DS} = 1$ V, for different $V_{TG}$. **d** Spiking threshold ($V_{ST}$), which we define as $V_{PSV}$ required to invoke a current spike, i.e., $I_{PSC} > I_{ST}$ as a function of $V_{TG}$, for different thresholding current, $I_{ST}$. As expected, the spiking threshold is higher for more negative $V_{TG}$ and higher $I_{ST}$. **e** An example of neuromorphic transducer comprising of Si PD and a parallel load resistor ($R_L$). The Si PD transduces light from the LED into photocurrent, which is converted to $V_{PSV}$ by $R_L$. **f** Current ($I_{PD}$) vs. voltage ($V_{PD}$) characteristics of the Si PD for different LED illuminance ($P_{LED}$). **g** Phototransduction characteristics of the Si PD and corresponding presynaptic voltage ($V_{PSV}$) applied to the neuromorphic encoder as a function of $P_{LED}$.

distribution with mean, $\mu_{TG} = -2.5$ V, and standard deviation $\sigma_{TG} = 0.8$ V as shown in Fig. 3b and the responses corresponding to each $V_{PSV}$ value with $I_{ST} = 500$ pA is displayed in Fig. 3c (see "Methods" section for discussion on current-sampling method). During each trial, $V_{TG}$ pulses were sampled for $N$ number of times (=32) and a total of 16 trials were conducted resulting in 512 sampling points for each $V_{PSV}$. See Supplementary Fig. 2 for the complete circuit used to obtain neural encoding for different illumination levels. Also, see Supplementary Movie 1 for real-time encoding of different LED intensities into stochastic spike trains. Figure 3d shows the encoding transfer function, i.e., mean firing rate (inverse of the mean interspike interval) as a function of $V_{PSV}$ (see Supplementary Fig. 3 for distribution of interspike interval). Clearly, the firing rate increases monotonically with increasing stimulus intensity, indicating that our biomimetic encoder is capable of rate-based encoding. Finally, Fig. 3e shows the encoding energy per spike ($E_{en}$) for rate-based encoding, computed based on Eq. (1). The monotonic increase in $E_{en}$ with increasing $V_{PSV}$ is consistent with increasing firing rate, i.e., more spiking in the postsynaptic neuron.

$$E_{en} = \frac{1}{N} \sum_{i=1}^{N} \left( \frac{1}{2} C_{TG} V_{TG,i}^2 + V_{PSC,i} V_{DS} t_p \right), \quad (1)$$

Typical energy consumption is around 1–5 pJ/spike. Note that the second term in Eq. (1) dominates in our demonstration since the first term contributes $\approx 100$ fJ. Therefore, one obvious way to

reduce the power consumption is through $V_{DS}$ scaling. Note that the neural encoder exploits subthreshold device characteristics and does not impose any requirement on the current device. Hence it is possible to operate the neural encoder with ultra-low $V_{DS}$. Another alternative to reduce the power dissipation is to increase the sampling rate i.e., reduce $t_p$. However, beyond a certain point, the first term will start to dominate, which can be scaled by scaling the oxide thickness to achieve encoding at scaled $V_{TG}$ values. Note that oxide thickness scaling increases $C_{TG}$, but the square term involving $V_{TG}$ will determine the energy scaling.

Another encoding principle found in sensory neurobiology is spike count-based encoding. For example, rats show remarkable texture discriminations using their facial whiskers. It is found that the trigeminal ganglion cells that innerte the sensory receptor from each whisker use spike count to distinguish the stimuli[33]. Similar spike count encoding is observed for frequency discrimination of vibrotactile stimuli in the primary somatosensory cortex of trained monkeys[34]. Figure 3f shows the $V_{TG}$ pulse profile used to achieve spike count-based encoding using our biomimetic encoder. In this case, the magnitude of $V_{TG}$ pulses increases with added zero mean Gaussian noise of standard deviation $\sigma_{TG} = 0.2$ V. Each trial consists of 32 pulses, and 16 trials were recorded for each $V_{PSV}$. The corresponding responses of the neural encoder are displayed in Fig. 3g. Figure 3h shows the encoding transfer function i.e., the mean spike count as a function of $V_{PSV}$ (see Supplementary Fig. 4 for total spike counts for all 16 trials for different $\sigma_{TG}$). Clearly, the mean spike count increases monotonically with increasing stimulus intensity, indicating that

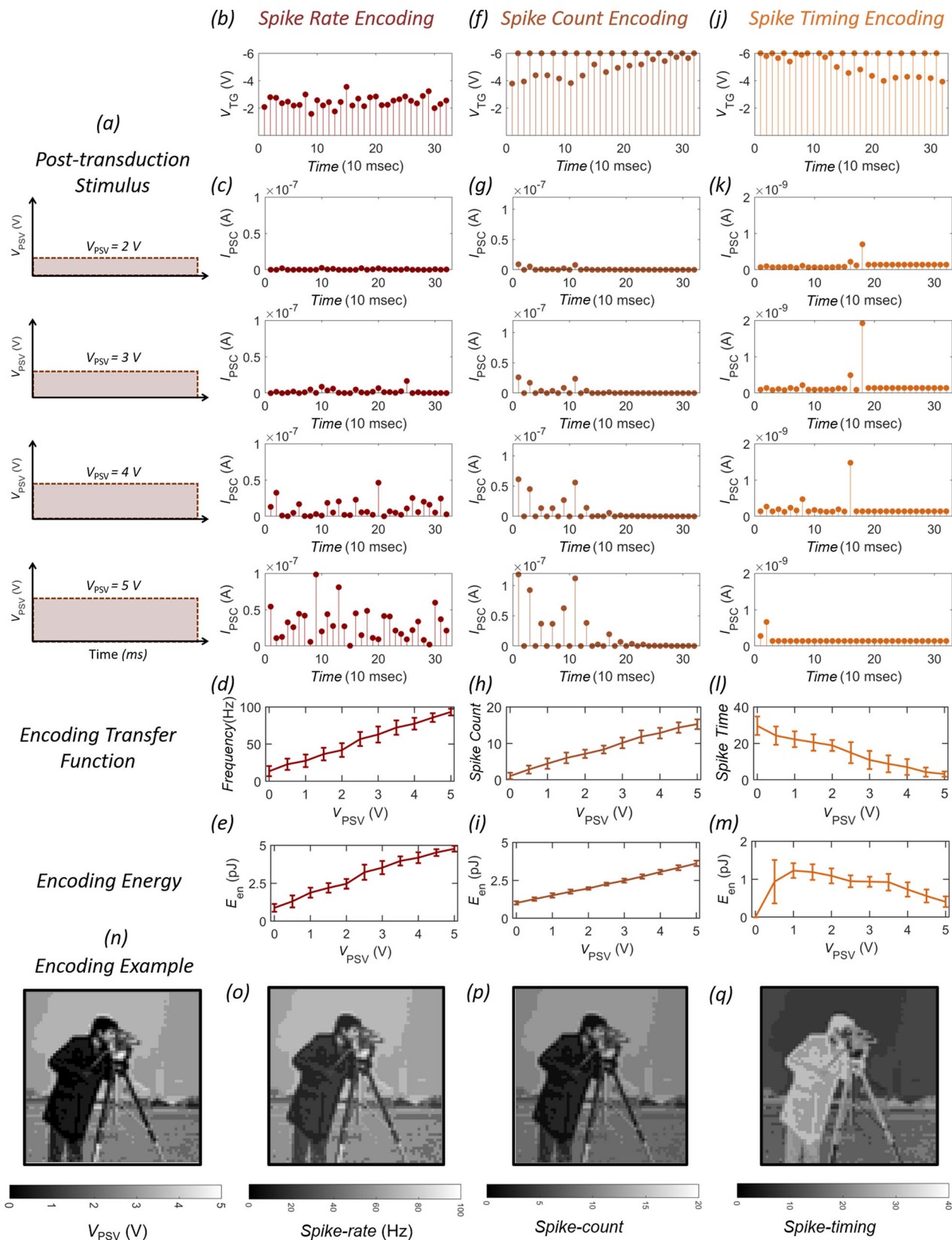

our biomimetic encoder is capable of spike count-based encoding. Note that, the implementation of spike count-based encoding does not necessarily require stochasticity, i.e., similar results could be obtained using $\sigma_{TG} = 0$ V. However, in the context of SNN, stochasticity can aid as hardware realization of integrate and fire (IF) neuron can be challenging. A more realistic neuron is leaky integrate and fire (LIF) neuron, where random spiking can compensate for the loss in information due to capacitive discharging between spikes. Figure 3i shows $E_{en}$ for spike count-based encoding, which shows a monotonic increase with $V_{PSV}$ since the spike count increases accordingly. The energy expenditure was found to be ≈1–3.5 pJ/spike.

**Fig. 3 Hardware realization of neural encoding algorithms. a** Analog $V_{PSV}$ values obtained for different LED illuminations. **b** Spike rate-based encoding using $V_{TG}$ pulses that are randomly sampled from a Gaussian distribution with mean, $\mu_{TG} = -2.5$ V, and standard deviation $\sigma_{TG} = 0.8$ V. **c** Corresponding $I_{PSC}$ for each $V_{PSV}$ in **a**. During each trial, $V_{TG}$ pulses were sampled for $N = 32$ number of times and a total of 16 trials were conducted resulting in 512 sampling points for each $V_{PSV}$. See Supplementary Movie 1 for real-time encoding of different LED intensities into stochastic spike trains. **d** Mean firing rate (frequency) as a function of $V_{PSV}$. **e** Encoding energy ($E_{en}$) for rate-based encoding. **f** Spike count-based encoding. Here, the magnitude of $V_{TG}$ pulses increase with added zero mean Gaussian noise of standard deviation $\sigma_{TG} = 0.2$ V. Each trial consists of 32 pulses and 16 trials are recorded for each $V_{PSV}$. **g** Corresponding responses of the neural encoder. **h** Mean spike count as a function of $V_{PSV}$. **i** $E_{en}$ for spike count-based encoding. **j** Spike timing-based encoding. Here, the magnitude of $V_{TG}$ pulses decrease over time with added zero mean Gaussian noise of standard deviation $\sigma_{TG} = 0.2$ V. Each trial consists of 32 pulses for each $V_{PSV}$. **k** Corresponding responses of the neural encoder. A sense amplifier is used to sense the arrival of the first spike triggering the deactivation of the $V_{TG}$ pulse sampling for the rest of the trial. **l** Mean spike-timing as a function of $V_{PSV}$. **m** $E_{en}$ for spike timing-based encoding. Error bars in **d, e, h, i, l, m** represent the variation (standard deviation) across trials. **n** Original, **o** spike-rate encoded, **p** spike-count encoded, and **q** spike-timing encoded Cameraman image. The pixel values ranging from 0 to 255 were mapped linearly to the $V_{PSV}$ range of 0 to 5 V. Supplementary movie files 2 show the time evolution of encoding of the Cameraman images for rate-based, count-based, and timing-based encoding.

Whereas, rate-based encoding and spike count-based encoding are the most broadly accepted view of neural computation, these approaches ignore the information possibly contained in the exact timing of the spikes. In fact, recent studies suggest that a straightforward firing rate or spike count-based encoding may be too simplistic to describe brain activity in its entirety. For example, neurophysiological experiments show that visual neurons in rhesus monkeys can recognize faces within ≈80–160 ms[35]. Anatomically, it involves more than ten synaptic stages between the photoreceptors of the retina and visually responsive neurons in the temporal cortex implying that each layer has, on average, only 10 ms of processing time. Since the firing rates of cortical neurons are in the range 0–100 spikes per second, a neuron in any given layer can only generate one spike before neurons in the next layer have to respond. This puts severe constraints on the way information is encoded in visual pathways. Firing-rate or spike count-based encoding seems inappropriate and evidence suggests that analog information is encoded by the relative arrival times of spikes[36–39]. Such an encoding scheme also referred to as the spike timing-based encoding, not only allows very rapid information processing but also offers tremendous energy benefits for future SNNs. Figure 3j shows the $V_{TG}$ pulse profile used for achieving spike timing-based encoding using our biomimetic encoder. In this case, the magnitude of $V_{TG}$ pulses decreases over time with added zero-mean Gaussian noise of standard deviation $\sigma_{TG} = 0.2$ V. Each trial consists of 32 pulses for each $V_{PSV}$. The corresponding responses of the neural encoder are displayed in Fig. 3k. A sense amplifier is used to sense the arrival of the first spike triggering the deactivation of the $V_{TG}$ pulse sampling for the rest of the trial. Figure 3l shows the encoding transfer function i.e., mean spike-timing as a function of $V_{PSV}$ (see Supplementary Fig. 5 for distribution of spike timing over multiple trials). Clearly, high-intensity stimuli invoke early spiking and vice versa indicating that our biomimetic encoder is capable of spike timing-based encoding as well. Note that, the implementation of spike timing-based encoding does not require stochasticity, i.e., similar results could be obtained by using $\sigma_{TG} = 0$ V. However, the flexibility of noise adjustment makes our neural encoder more bio-realistic. Figure 3m shows $E_{en}$ for spike timing-based encoding. Unlike rate-based and count-based encoding, timing-based encoding shows a monotonic decrease with increasing $V_{PSV}$. This is owing to the fact the spiking occurs earlier for higher $V_{PSV}$ deactivating the encoder and minimizing the energy consumption per spike. The fact that the encoding energy can be significantly lower for timing-based encoding compared to rate-based or count-based encoding is appealing for ultra-low-power neuromorphic computing using SNN (See Supplementary Note 2 showing the comparison of our neural encoder with other types of spike encoders).

Finally, Fig. 3n–q, respectively, shows the original Cameraman image and the corresponding spike rate, spike count, and spike timing-based encoding. The pixel values ranging from 0 to 255 were mapped linearly to the $V_{PSV}$ range of 0–5 V (see the "Methods" section for details on image encoding). Clearly, the Cameraman image is accurately encoded, irrespective of the encoding algorithm. Note that the contrast of the image in Fig. 3q is reversed compared to the original image, which is expected for spike-time based encoding since the higher pixel values should spike earlier than the lower pixel values. Supplementary movie files 2 show the time evolution of encoded images over time for rate-based, count-based, and timing-based encoding. Supplementary Fig. 6 shows the time evolution of the correlation coefficient (CC) between the original image and the encoded image. The CC reaches ≈1 at the end of encoding for all three encoding algorithms.

**Dynamic range and encoding precision for rate-based encoding.** Now, we focus on two key aspects of neural encoding, namely, dynamic range and encoding precision. A high dynamic range (HDR) allows neurons to respond to more extreme stimuli. For example, photoreceptors in human eyes can identify objects in starlight as well as in bright sunlight despite of illumination levels differing by ≈9 orders of magnitude, i.e., over a dynamic range of 90 dB[40]. Similarly, the dynamic range of human hearing is roughly 140 dB[41]. However, HDR does not necessarily guarantee high precision (HP). For example, a whisper cannot be heard in loud surroundings. Similarly, eyes take time to adapt to different illumination levels. In fact, most sensory neurons adjust their spike encoding based on the environment[42–44]. Figure 4a–f shows how our biomimetic encoder achieves similar functionality by adjusting $\sigma_{TG}$ and $\mu_{TG}$ of the Gaussian distribution used for sampling $V_{TG}$ as well as and the thresholding current, $I_{ST}$ for the spike rate-based encoding algorithm presented earlier. For numerical simulations, we have used the virtual source (VS) model described elsewhere[22,23]. Clearly, HDR can be achieved by using higher values of $\sigma_{TG}$, whereas smaller values of $\sigma_{TG}$ allow HP (Fig. 4a, b). This is because the encoding transfer function follows the cumulative probability distribution of a random Gaussian variable since for a given $V_{PSV}$ stimulus, there will always be a postsynaptic spike if the magnitude of the $V_{TG}$ pulse is more positive than the one corresponding to the spiking threshold, $V_{ST}$, as shown in Fig. 2c. For a higher value of $\sigma_{TG}$, the cumulative probability distribution follows a linear trend allowing a larger $V_{PSV}$ range to be encoded, whereas a lower value of $\sigma_{TG}$ results in a non-linear cumulative probability distribution that restricts the encoding range but improves the encoding precision. However, both feats cannot be achieved at the same time by adjusting $\sigma_{TG}$. However, by adjusting $\mu_{TG}$ it is possible to achieve HP for different ranges of stimulus intensity similar to the

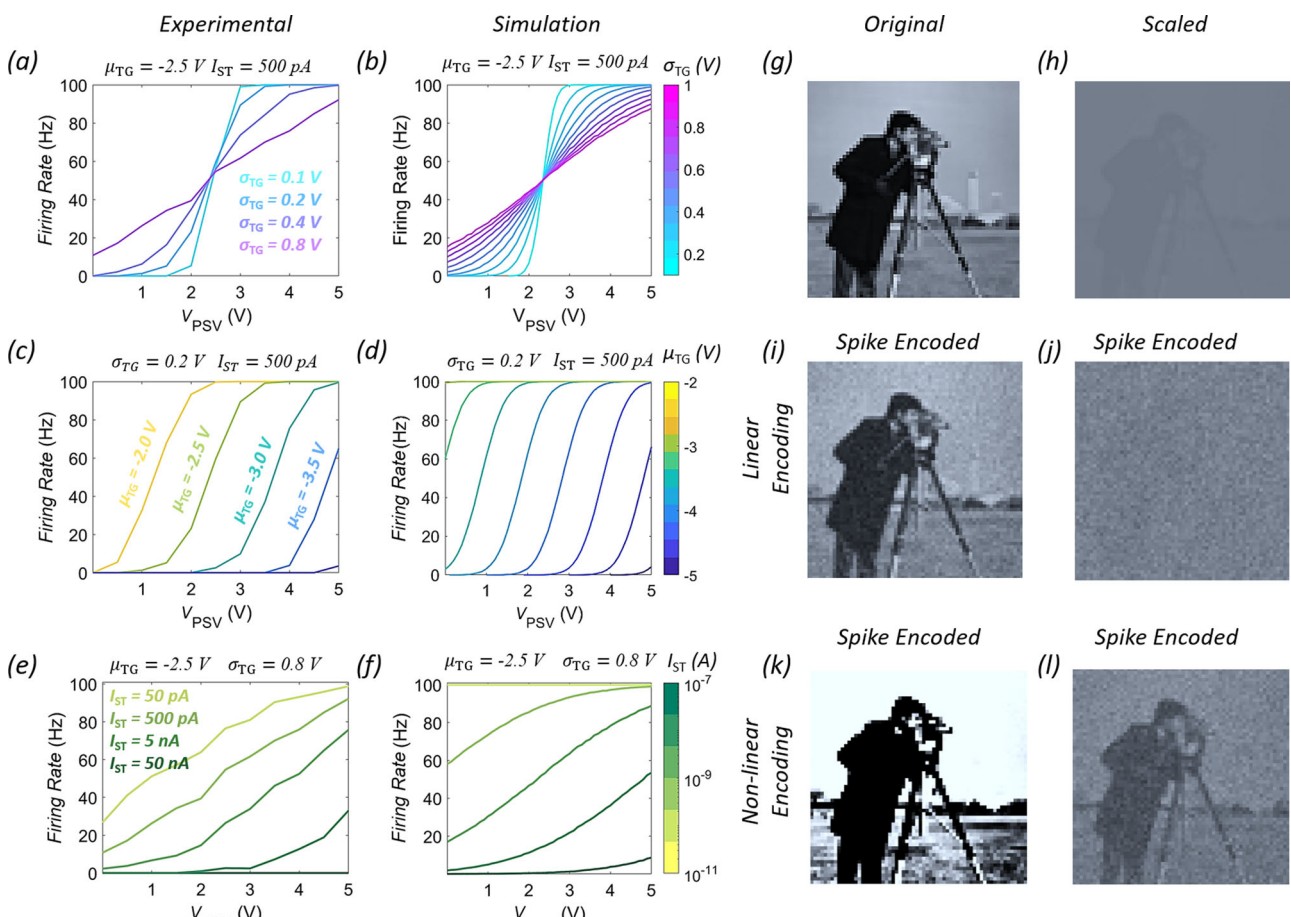

**Fig. 4 Dynamic range and encoding precision for rate-based encoding. a** Experimental and **b** virtual source (VS) model-based simulation of firing rate as a function of $V_{PSV}$, for different $\sigma_{TG}$. of the Gaussian distribution used for sampling $V_{TG}$. A high dynamic range (HDR) can be achieved by using higher values of $\sigma_{TG}$, whereas smalle. r values of $\sigma_{TG}$. allow high precision (HP). However, both feats cannot be achieved at the same time by adjusting $\sigma_{TG}$. **c** Experimental and **d** VS model-based simulation of firing rate as a function of, $V_{PSV}$, for different mean ($\mu_{TG}$) of the Gaussian distribution used for sampling $V_{TG}$. By adjusting $\mu_{TG}$ it possible to achieve HP for different ranges of stimulus intensity like the sensory neurons. **e** Experimental and **f** VS model-based simulation also show that the firing rate can be tuned by adjusting $I_{ST}$. Lower values of $I_{ST}$ allow more spiking events for any given $V_{PSV}$, whereas, higher values of $I_{ST}$ restrict. s spiking even for higher $V_{PSV}$. **g** Original and **h** scaled Cameraman images. Corresponding spike rate-based **i, j** linear and **k, l** non-linear encoding using our biomimetic encoder. The original Cameraman image necessitates linear encoding since the pixel values have large dynamic range, whereas, the scaled Cameraman image is better encoded using high precision non-linear encoding.

sensory neurons (Fig. 4c, d). The spiking rate can also be tuned by adjusting the $I_{ST}$ (Fig. 4e, f). Lower values of $I_{ST}$ allow more spiking events for any given $V_{PSV}$, whereas, higher values of $I_{ST}$ restrict spiking even for higher $V_{PSV}$. Figure 4g and h show the original and scaled Cameraman images, and Fig. 4i–l, respectively, show the corresponding spike rate-based linear and non-linear encoding using our biomimetic encoder. The original Cameraman image necessitates linear encoding since the pixel values have a large dynamic range, whereas, the scaled Camera-man image is better encoded using high precision non-linear encoding.

**MNIST digit classification using our neural encoder device.** Finally, we exploit our biomimetic encoder for encoding MNIST data set on digit-classification into spike trains and infer using a trained SNN. For training the SNN we have used an approach described by Sengupta et al.[45]. This approach overcomes the lower accuracy of unsupervised learning rules such as the spike-time dependent plasticity (STDP) used for training SNNs[46–50]. The lower accuracy is due to the lack of efficient algorithms to make use of the spiking neurons. To bridge this gap, ANN-SNN

conversion schemes are used, where an ANN is trained using the traditional back-propagation algorithm, followed by the conversion of the ANN to SNN[45,51,52]. This approach yields higher inference accuracy owing to near-lossless ANN-SNN conversion[45]. Here, we train a fully connected two-layered arti-ficial neural network with 100 neurons in the hidden layer and 10 neurons in the output layer for digit-classification using the MNIST dataset as shown in architecture in Fig. 5a. MNIST dataset with a size of $28 \times 28$ pixels is flattened to obtain 784 pixels, which is fed to the input layer. The ten output neurons correspond to digits from 0 to 9. During training, for every input image, the network is trained through gradient descent to ensure that the output matches the expected label. Here, the ANN is trained with a learning rate of 0.0001 to ensure high convergence accuracy. Further, the following restrictions are incorporated while training the ANN to allow smooth ANN-SNN transition: rectified linear unit (ReLU) is used as the activation function due to its functional equivalence to IF spiking neuron used in SNN, bias terms are eliminated to ensure a smaller parameter space which enables easier ANN-SNN conversion, and no regulariza-tion is used. Sixty thousand images from the MNIST data set were used to train the ANN to achieve a training accuracy of 91.5%

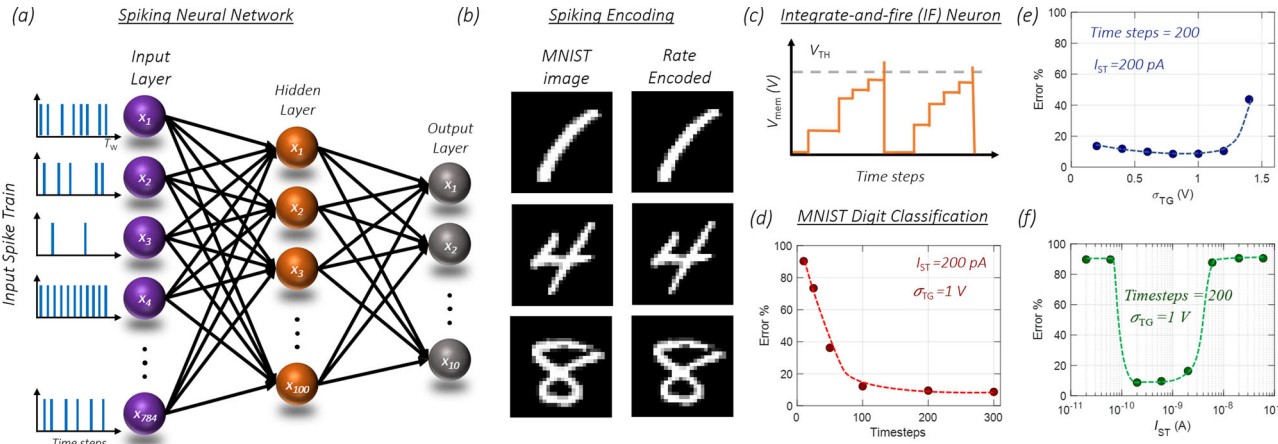

**Fig. 5 Encoding of MNIST data for digit classification using SNN. a** A fully connected two-layered trained ANN with 100 neurons in the hidden layer and 10 neurons in the output layer for MNIST digit-classification is transformed into SNN. **b** Examples of spike encoded digits. First, the pixel intensity values ranging from 0 to 255 are mapped on to $V_{PSV}$ values in the range of 0–5 V. Next, we recorded $I_{PSC}$ for each $V_{PSV}$ by applying stochastic $V_{TG}$ pulses with $I_{ST}$ = 200 pA to generate binary spike trains ($X$) in time. We adopt rate-based encoding by applying $V_{TG}$ pulses with the pulse magnitude determined using a random Gaussian distribution with $\mu_{TG} = -5.5$ V, and $\sigma_{TG} = 1$ V. **c** Characteristics of IF neuron which is substituted for ReLU for ANN-SNN conversion. In the IF neuron when the membrane potential crosses a certain threshold ($V_{TH}$), the neuron spikes, propagating spike to the next layer, and it resets back to its resting potential which is set as zero. **d** Inference error versus the number of timesteps for classifying a set of 10,000 test images. Increasing the number of timesteps is important to allow sufficient firing, to effectively encode the pixel intensities. But even with 200 timesteps, we achieve a low error of 9.5%. Dependence of inference accuracy on **e** $\sigma_{TG}$, and **f** $I_{ST}$. The minimum error of ≈8.6% is achieved at $\sigma_{TG} = 0.8$ V. For lower $\sigma_{TG}$, the dynamic range is insufficient to capture the variation in pixel intensities, whereas, for very high $\sigma_{TG}$, there is insufficient difference between the firing rates corresponding to different pixel intensities resulting in larger errors. Similarly, the minimum error of 8.6% is obtained for $I_{ST} = 200$ pA. For higher $I_{ST}$, the spiking is minimal resulting in an inadequate representation of image pixels, whereas, for lower $I_{ST}$, any pixel intensity results in excessive firing.

over 100 epochs. Following this, a testing accuracy of 92.7% was achieved using the remaining 10,000 images.

As discussed, SNNs use binary spikes in time which are representative of the action potential in BNNs. This requires the conversion of the analog image pixel intensities to digital spike trains. To accomplish the analog to spike conversion using our biomimetic encoder, first, the pixel intensity values ranging from 0 to 255 are mapped onto $V_{PSV}$ range of 0–5 V. Next, we record $I_{PSC}$, for $V_{PSV}$ values corresponding to each pixel over a time window, $T_W$, by applying stochastic $V_{TG}$ pulses. Each $I_{PSC}$ value subsequently undergoes a thresholding function with $I_{ST}$ to generate binary $X$ in time. We adopt rate-based encoding by applying $V_{TG}$ pulses with the pulse magnitude determined using a random Gaussian distribution, as described earlier. Figure 5b shows examples of spike encoded digit using $\mu_{TG} = -5.5$ V, and $\sigma_{TG} = 1$ for the Gaussian distribution for the $V_{TG}$ pulses, and $I_{ST} = 200$ pA. The resultant $X$ is fed into the SNN, as shown in the input layer of Fig. 5a. For ANN-SNN conversion, ReLU activation functions are replaced by IF neuron as shown in Fig. 5c following Eq. (2).

$$V_{mean}(t+1) = V_{mean}(t) + \sum w \cdot X(t). \quad (2)$$

Here, the IF neuron is represented as the function of timesteps ($t$). $V_{mean}(t)$ is the membrane potential, and $w$ denotes the weights obtained from the trained ANN. In the IF neuron when the membrane potential crosses a certain threshold ($V_{th}$), the neuron spikes, propagating spike to the next layer, and it resets back to its resting potential which is set as zero. To optimize the IF neuron threshold, threshold-balancing is used to set the threshold as the maximum neuron activation for the corresponding layer obtained by the dot product of the weights and spike-train at an instance $t$[45]. The SNN is used to classify the set of 10,000 test images. Figure 5d shows the inference error versus the number of timesteps. Increasing the number of timesteps is important to allow sufficient firing, to effectively encode the pixel

intensities. But remarkably, even with 200 timesteps, we achieve a low error of 9.5%. This is further improved as the timesteps are increased with the minimum error of 8.6% at 300 timesteps. Hence a maximum accuracy of 91.4% is achieved when the SNN is simulated with our biomimetic neural encoder. Additionally, similar test accuracies are obtained from both ANN and SNN, indicating a successful ANN-SNN transformation with a minimal loss of 1.3%.

Finally, we explore the dependence of inference accuracy on the dynamic range and the firing rate, parameters that can be adjusted in our biomimetic spike encoder by adjusting $\sigma_{TG}$ and $I_{ST}$, respectively. As shown in Fig. 5e, minimum error of ≈8.6% is achieved at $\sigma_{TG}$ of 0.8 and 1, with higher errors for lower and higher $\sigma_{TG}$. As described earlier in Fig. 4a, b, for lower $\sigma_{TG}$, the dynamic range is low to capture the variation in pixel intensities, whereas for very high $\sigma_{TG}$, there is insufficient difference between $t$ firing rates corresponding to different pixel intensities resulting in a large error. A similar non-monotonic trend is seen in inference error with respect to $I_{ST}$ in Fig. 5f, with the minimum error of 8.6% obtained at 200 pA. For higher $I_{ST}$, the spiking is minimal resulting in an inadequate representation of image pixels, while for lower $I_{ST}$, any pixel intensity results in excessive firing as seen in Fig. 4e, f. Nevertheless, by optimizing these parameters it is possible to ensure efficient encoding of the MNIST images, and thereby achieve a maximum accuracy of 91.4%.

## Discussion

In conclusion, we have developed a neural encoder based on a dual gated $MoS_2$ FET with a stochastic sampling terminal capable of encoding analog signals into spike trains. We also implemented three encoding algorithms, namely, spike rate-based encoding, spike count-based encoding, and spike timing-based encoding found in sensory neurobiology. As a prototype demonstration, we

show the direct conversion of analog light intensities to corresponding rate-based spike trains analogous to phototransduction mechanisms in visual pathways. We also show frugal encoding energy expenditure in the range of few pico Joules per spike. Our biomimetic encoder also allows flexibility in terms of adjusting the encoding range and encoding precision, two key features found in biological sensory transduction to enable seamless adaption to different environmental conditions. Finally, we encoded the MNIST data set for digit classification using our spike encoder and achieved an inference accuracy of 91.4% by using a trained SNN. In brief, our demonstration of the biomimetic neural encoder is a leap forward towards achieving energy-efficient and bio-realistic neuromorphic hardware.

## Methods

**Device fabrication**. The dual-gated devices were fabricated using micromechanically exfoliated $MoS_2$ flakes on 285 nm thermally grown $SiO_2$ substrates with highly doped-Si as the back-gate electrode. The source/drain contacts were defined using electron-beam lithography (Vistec EBPG5200). Ni (40 nm) followed by Au (30 nm) were deposited using electron-beam evaporation for the contacts. For fabricating the top-gate, hydrogen silsesquioxane (HSQ) was used as the dielectric. It was deposited by spin coating 6% HSQ in methyl isobutyl ketone (MIBK) (Dow Corning XR-1541-006) at 4000 rpm for 45 s and baked at 80 °C for 4 min. The HSQ was patterned using an e-beam dose of 2000 μC/cm$^2$ and was developed at room temperature using 25% tetramethylammonium hydroxide (TMAH) for 30 s following a 90 s rinse in deionized water (DI). Next, it was cured in the air at 180 °C and then 250 °C for 2 min and 3 min, respectively. The top-gate electrode was patterned using electron-beam lithography followed by the deposition of Ni/Au using electron-beam evaporation as the contact.

**Device measurements**. Electrical characterization was performed at room temperature in high vacuum (≈$10^{-6}$ Torr) on a Lake Shore CRX-VF probe station and using a Keysight B1500A parameter analyzer. We observed none to minimal hysteresis in the device characteristics for both top-gate and back-gate sweeps indicating high quality of $MoS_2/SiO_2$ and $MoS_2/HSQ$ interfaces (See Supplementary Fig. 7). For current sampling, when a sampling delay is set, (for example, $T = 10$ ms) the tool determines a time width (W) for integration based on the current value i.e., lower current values need larger $W$ and vice versa. If $W < T$ the tool measures the current at any point within the delay period. Else if $W > T$, irrespective of the time delay, the tool measures the current every time width (W) it sets for the current value range. The 10 ms delay we chose is large enough for the current value ranges we are operating to integrate. So, the tool will measure at any point within the 10 ms delay.

**Image encoding**. Note that the gray scale pixel values in an 8-bit cameraman image range from 0–255, which are mapped to $V_{PSV}$ range of 0–5 V. This would require a $V_{PSV}$ precision of $5/255 = 0.0196 ≈ 0.02$ V. However, our experimental $V_{PSV}$ step size was 0.5 V. Therefore, we quantized the cameraman image into 11 distinct levels and correlate those levels to $V_{PSV} = 0, 0.5, 1.0, 1.5, 2.0, 2.5, 3.0, 3.5, 4.0, 4.5,$ and 5.0 V. Note that for spike-rate based encoding, our maximum frequency is 100 Hz and the average standard deviation for the encoded frequencies i.e., the encoding error bar is ≈10 Hz, allowing distinct encoding of 11 levels. One way to improve the mapping precision for spike-rate based encoding is to increase the maximum encoding frequency to 1000 Hz through faster sampling if the encoding error bar remains unaltered. Similarly, for spike-count based encoding, the maximum count is 16 and the average standard deviation for the encoded count or the encoding error bar is ≈1.5, again allowing the distinct encoding of 11 levels. Finally, for spike-timing based encoding, the maximum number of time-steps is 30 and the average standard deviation for the encoded time-step or the encoding error bar is ≈4, allowing the distinct encoding of eight levels. The mapping precision for spike-count and spike-timing based encoding can be increased by increasing the total number of sampling points.

## Data availability

The datasets generated during and/or analyzed during the current study are available from the corresponding author on reasonable request.

## Code availability

The codes used for plotting the data are available from the corresponding authors on reasonable request.

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

## Acknowledgements

The work was supported by Army Research Office (ARO) through Contract Number W911NF1920338. Figure 1a was designed using resources from Freepik.com "image: Freepik.com".

## Author contributions

Saptarshi Das and S.S. conceived the idea and designed the experiments. Saptarshi Das wrote the paper. Sarbashis Das fabricated the devices. A.S. performed SNN simulations. Saptarshi Das, S.S., A.S., A.O., and Sarbashis Das analyzed the data, discussed the results, agreed on their implications, and contributed to the preparation of the manuscript.

## Competing interests

The authors declare no competing interests.
