## [Peer Review File · Nature Communications]

REVIEWER COMMENTS

Reviewer #1 (Remarks to the Author):

The manuscript entitled "A Biomimetic Neural Encoder for Spiking Neural Network" by Shiva Subbulakshmi Radhakrishnan et al. reported a neural encoder based on dual gated MoS₂ field effect transistor, which can encode analog signals into stochastic spike trains. Three kinds of neural encoding algorithms were demonstrated by using the neural encoder to encode analog light intensities, and good encoding performance was achieved. The authors showed a classification accuracy of 91.4% for MNIST dataset by using a trained SNN. The energy consumption was found to be ~200 pJ/spike. This work may offer a route to the design of hardware encoder for spiking neural networks. This paper in my opinion is suitable for the readership of Nature Communications. However, I think there are some issues to be resolved.

- The authors need to list a table to compare the device performance between other types of spike encoders and the present one.
- The energy consumption for memristors in crossbar array seems to be ~pJ level. How to further reduce energy consumption for this spike encoder?
- The authors designed field effect transistors by utilizing multilayer MoS₂ in this work but monolayer MoS₂ in the previous work (Adv. Electron. Mater. 2019). Both devices have similar configurations and transfer characteristics. Why did the authors utilize multilayer MoS₂ for spike encoder?
- In line 15 page 9, "During each trial, VTG pulses were sampled, N= 512 times for each VPSV." The authors should explain the meaning of the letter "N".
- There is no text description about z-axis in extended data 3 and extended data 5.

Reviewer #2 (Remarks to the Author):

This paper demonstrates MoS₂-based dual gated field effect transistors (FETs) for neural encoder application. Analog signals are applied to the back-gate terminal and encoded spikes are sampled out through top-gate voltage pulses. Various neural encoding algorithms are achieved. In addition, the effect of top-gate pulse distribution to the dynamic range and precision are analysed in terms of the quality of encoded images.

The integration of a light emitting diode (LED), a Si photodiode (PD), a MoS₂ transistor and a simulated spiking neural network to emulate a spiking neuromorphic system is interesting. The role of each element in this system is explained clearly, such as sensor, encoder and processor.

Obvious relationship between encoded spike trains and input analog signals are achieved in this work using different encoding algorithms. However, many details in terms of encoding methods and device characteristics in this manuscript need further explanation.

1. The authors need to explain the reason why they choose Ni as the contact metal. What are the possible effects if they use other metal for contact, such as Ti?
2. The FET transfer curve data in Figure 2c is not consistent with previous work.[1] In that work, the authors use 100nm SiO₂ back-gate and a range of 50 V is required to fully turn on the transistor. In this work, the authors use even thicker oxide layer of 285 nm, which should have less gate control to the channel. However, a range of only 14 V from back-gate is enough to fully turn on the transistor.
3. The authors claim their top-gate and back-gate can be interchanged. However, in Extended data 1d, the on/off ratio using top-gate is much lower than that using back-gate. The authors need to explain this. In addition, the reason for choosing top-gate as the presynaptic terminal (VPSV) needs to be

explained. What are the disadvantages for using back-gate as VPSV?

4. In Figure 2g, the choice of load resistor (RL) and VPSV range should be explained. Why the authors choose RL to achieve range of 0V-5V for VPSV instead of other ranges?

5. In Figure 2f and 2g, the authors use different range of LED power. One is 0-200mW, and another is 0-80mW. The author should make the range the same. Otherwise, it is difficult to compare the IPD in these two figures and check whether they are consistent.

6. In Figure 3c, 3g and 3k, the authors need to explain how to define the spike, and what the IST used in each encoding algorithm.

7. In Figure 3c, 3g and 3k, the authors use a 10ms VTG pulse for current sampling. The authors need to explain how they read the current during this 10ms period, by taking average of current, or choosing any point in this period, or other methods?

8. For Figure 3d, 3e, 3h, 3i, 3l and 3m, the authors should also show the standard deviation (error bar) of each point. For example, the authors take the mean spike interval to plot Figure 3d, but they also need to show the standard deviation (error bar) of the spike interval under each VPSV. Otherwise this trend line in Figure 3d is not very convincing.

9. In Figure 3d, 3e, 3h, 3j, 3l and 3m, the authors need to explain how they get the dashed lines. What are the functions they used to fit the points? In addition, the authors need to explain why Figure 3i and 3m do not follow linear relationship.

10. In Figure 3n-q, the authors need to explain how the images are encoded in detail. (i) It seems that the authors only have results at VPSV range of 0 to 5V with step of 0.5V. If pixel values are linearly mapped to VPSV range of 0 to 5V, what are the spike rate, spike count and spike timing at other VPSV values? For example, $VPSV = 0.02$ V. (ii) What do the pixel values in Figure 3o-q represent for? If the pixel values in the encoded images refer to spike rate, spike count and spike timing, Figure 3q should be an inverse image, since larger VPSV leads to small spike time value (Figure 3l).

11. In Extended data 6, the authors use Lena image to compare the correlation coefficient (CC), which is not consistent with the main text.

12. The authors should list a table for all the related parameters they used for the encoding transfer function (especially for Figure 4a, 4c, 4e), including VTG average, VTG standard deviation and IST. Otherwise, it is difficult to compare whether Figure 4a, 4c and 4e are consistent or not.

13. Is there any full lobe of the transfer curve for both top-gate and back-gate? The authors need to show that there is no hysteresis in this dual gated transistor. Otherwise, hysteresis can significantly influence the sampling results depending on the previous pulse amplitude.

14. What is the thickness of MoS2 used in this work? Will the thickness affect the dual gate control?[2]

References

[1] J. R. Nasr, S. Das, *Adv. Electron. Mater.* 2019, 5, 1800888.

[2] C. Liu, H. Chen, X. Hou, H. Zhang, J. Han, Y.-G. Jiang, X. Zeng, D. W. Zhang, P. Zhou, *Nanotechnol.* 2019, 14, 662.

Reviewer #1 (Remarks to the Author):

The manuscript entitled “A Biomimetic Neural Encoder for Spiking Neural Network” by Shiva Subbulakshmi Radhakrishnan et al. reported a neural encoder based on dual gated MoS2 field effect transistor, which can encode analog signals into stochastic spike trains. Three kinds of neural encoding algorithms were demonstrated by using the neural encoder to encode analog light intensities, and good encoding performance was achieved. The authors showed a classification accuracy of 91.4% for MNIST dataset by using a trained SNN. The energy consumption was found to be ~200 pJ/spike. This work may offer a route to the design of hardware encoder for spiking neural networks. This paper in my opinion is suitable for the readership of Nature Communications. However, I think there are some issues to be resolved.

We would like to thank the reviewer for their appreciation of our work. We have resolved the issues raised by the reviewer in the revised manuscript as described below.

The authors need to list a table to compare the device performance between other types of spike encoders and the present one.

This is an excellent suggestion.

We have added the following benchmarking table as *Supplementary Note 2*.

Reference	Technology	Hardware Encoding	Spike Encoding Type			Energy per synaptic event
			Rate	Count	Temporal	
[1]	Memristor	✓	✗	✗	✓	25 pJ
[2]	CMOS 180nm	✗	✓	✗	✓	225 fJ
[3]	CMOS 65nm	✓	✓	✗	✗	40 fJ
[4]	MTJ	✗	✓	✗	✗	48 fJ
[5]	PCM	✗	✓	✗	✗	-
[6]	FPGA	✓	✗	✗	✓	21 nJ
[7]	CBRAM	✗	✗	✗	✓	-
This work	2D FET	✓	✓	✓	✓	< 3 pJ

The energy consumption for memristors in crossbar array seems to be ~pJ level. How to further reduce energy consumption for this spike encoder?

The reviewer has a valid concern. We have updated our energy consumption plots in Fig. 3e,3i & 3m, normalized to per spike to make a fair comparison with the existing work. Our average energy consumption is around 1-3 pJ which is on par with the memristor technology. Also, we are operating our device in the subthreshold regime, where the device current is independent of the applied drain bias, V_{DS} , as long as $V_{DS} > k_B T/q \approx 25$ meV. In fact, since the neural encoder does not impose any requirement for the device current it is possible to operate the neural encoder at ultra-low V_{DS} . In the present work we have used $V_{DS} = 1.0$ V. Therefore, one obvious way to reduce the power consumption is through V_{DS} scaling following Eq. 1. Note that the second term in Eq. 1 dominates in our demonstration (the first term contributes ~ 100 fJ).

$$E_{en} = \frac{1}{N} \sum_{i=1}^N \left(\frac{1}{2} C_{TG} V_{TG,i}^2 + I_{PSC,i} V_{DS} t_P \right) \quad [1]$$

Another alternative to reduce the power dissipation is to increase the sampling rate i.e. reduce t_P . However, beyond a certain point the first term will start dominating, which can be scaled by scaling the oxide thickness to achieve encoding at scaled V_{TG} . Note that oxide thickness scaling increases C_{TG} but the square term involving V_{TG} will determine the energy scaling.

We have added the above discussion in the revised manuscript.

The authors designed field effect transistors by utilizing multilayer MoS2 in this work but monolayer MoS2 in the previous work (Adv. Electron. Mater. 2019). Both devices have similar configurations and transfer characteristics. Why did the authors utilize multilayer MoS2 for spike encoder?

We thank the reviewer for bringing up this point. The work, Adv. Electron. Mater. 2019, demonstrates that it is possible to fabricate dual-gated FETs using HSQ in a seamless, scalable, and universally applicable method. We have adopted the same fabrication technique for our neural encoder device used in this manuscript. The purpose of our current work is to demonstrate a proof-

of-concept solid-state device capable of encoding analog signals into stochastic spike trains using different neural encoding schemes. With the device characteristics being similar for monolayer versus multilayer devices and the focus being on the neural encoding functionality, we did not focus on optimizing the semiconductor layer itself. It is possible to demonstrate the same functionality on monolayer FETs using the same dual-gate stack.

In line 15 page 9, “During each trial, VTG pulses were sampled, N= 512 times for each VPSV.” The authors should explain the meaning of the letter “N”.

We have defined the meaning of the letter “N” in the revised manuscript.

There is no text description about z-axis in extended data 3 and extended data 5.

We have added the text description about z-axis in extended data 3 (now *Supplementary Figure 3*) and extended data 5 (now *Supplementary Figure 5*).

Reviewer #2 (Remarks to the Author):

This paper demonstrates MoS₂-based dual gated field effect transistors (FETs) for neural encoder application. Analog signals are applied to the back-gate terminal and encoded spikes are sampled out through top-gate voltage pulses. Various neural encoding algorithms are achieved. In addition, the effect of top-gate pulse distribution to the dynamic range and precision are analyzed in terms of the quality of encoded images.

The integration of a light emitting diode (LED), a Si photodiode (PD), a MoS₂ transistor and a simulated spiking neural network to emulate a spiking neuromorphic system is interesting. The role of each element in this system is explained clearly, such as sensor, encoder, and processor.

Obvious relationship between encoded spike trains and input analog signals are achieved in this work using different encoding algorithms. However, many details in terms of encoding methods and device characteristics in this manuscript need further explanation.

We would like to thank the reviewer for their appreciation of our work. We have provided additional details in terms of encoding methods and device characteristics in the revised manuscript following reviewers comments as indicated below.

1. The authors need to explain the reason why they choose Ni as the contact metal. What are the possible effects if they use other metal for contact, such as Ti?

The reviewer has brought up an important point regarding the choice of nickel (Ni) as the contact metal. We agree that reduction in contact resistance through metal work-function engineering is an extremely relevant and widely studied topic in 2D FETs [8]. Earlier works have shown that irrespective of the metal work-function, the metal Fermi level pins closer to the conduction band of MoS₂ resulting in unipolar n-type device characteristics [9]. It is true that in spite of the phenomenon of Fermi level pinning lower work function metal like titanium (Ti) will offer lower Schottky barrier (SB) height (~50-100 meV) compared to high work-function metal like Ni (SB ~ 150-200 meV) resulting in higher FET on-currents for MoS₂ FETs with Ti. However, having higher on-current is not critical for the design of the neural encoder introduced in this work since the encoding is achieved by exploiting the subthreshold device characteristics ensuring low energy

expenditure. The impact of metal work-function on room-temperature subthreshold characteristics of ultra-thin body MoS₂ FETs is expected to be minimal since the difference in SB height between Ti and Ni contact is similar to the broadening of metal Fermi function ($\sim 2-4 k_B T/q \approx 50-100$ meV). An argument can be made in terms of relatively lower threshold voltage achievable by using Ti contact compared to Ni contact aiding low-power operation. However, note that threshold voltage is engineered seamlessly through electrostatic doping using the top-gate in our dual-gated MoS₂ FET. Therefore, the impact of metal contacts on neural encoding is expected to be insignificant. The choice of Ni as the metal contact is owing to less stringent demand on the vacuum condition during the electron beam evaporation compared to Ti, which requires ultra-high vacuum [10].

2. The FET transfer curve data in Figure 2c is not consistent with previous work [1]. In that work, the authors use 100nm SiO₂ back-gate and a range of 50 V is required to fully turn on the transistor. In this work, the authors use even thicker oxide layer of 285 nm, which should have less gate control to the channel. However, a range of only 14 V from back-gate is enough to fully turn on the transistor.

The reviewer has made an excellent observation. However, as explained below, the results presented in our current manuscript and previous report by Nasr *et al.* [11] are indeed consistent. First, we would like to point few important distinctions between the two works (Fig. R1):

1. The channel length (L) of the MoS₂ FETs used in the study by Nasr *et al.* [11] is 2 μm , whereas we have used $L = 1 \mu\text{m}$.
2. In the study by Nasr *et al.* [11], the transfer characteristics of the MoS₂ FETs are normalized to the channel width ($W = 2.15 \mu\text{m}$), whereas the device data presented in this work were not normalized to the channel width of $W = 2.84 \mu\text{m}$.
3. The electron field effect mobility (μ_{g_m}) values extracted from transconductance plateau were found to be $\mu_{g_m} = 8 \text{ cm}^2/\text{V}\cdot\text{s}$ for Nasr *et al.* (Fig. R1c) and $\mu_{g_m} \approx 12 \text{ cm}^2/\text{V}\cdot\text{s}$ (Fig. R1d) in the present work MoS₂.
4. The back-gate capacitance (C_{BG}) values, as pointed out by the reviewer, are $C_{\text{BG}} = 3.4 \times 10^{-4} \text{ F}/\text{m}^2$ for Nasr *et al.* (100 nm SiO₂) and $C_{\text{BG}} = 1.2 \times 10^{-4} \text{ F}/\text{m}^2$ in the present work (285 nm SiO₂).

5. Finally, the threshold voltage (V_{TH}) values extracted using constant current method at 100 nA/ μm were found to be $V_{TH} = -10$ V for Nasr *et al.* (Fig. R1a) and $V_{TH} = 4$ V in the present work (Fig. R1b). The difference in V_{TH} can be attributed to either fixed charges at the MoS₂/SiO₂ and MoS₂/HSQ interfaces arising due to chemically/physically adsorbed residual water/organic molecules from the lithography/curing processes or fixed charges inside SiO₂ and/or HSQ.

Figure R1. Back-gate transfer characteristics of multilayer MoS₂ FET with HSQ as the top-gate a) from Nasr *et al.* [11] with 100 nm SiO₂ as the back gate dielectric and b) this work with 285 nm SiO₂ as the back gate dielectric. Electron field effect mobility extracted using transconductance c) from Nasr *et al.* [11] and b) this work.

With these differences in mind, the expected overdrive voltage, i.e. ($V_{BG} - V_{TH}$), where V_{BG} is the back-gate voltage to achieve similar drive current (I_{ON}) values in the MoS₂ FETs can be derived from Eq. 1, which is valid for linear regime of the device operation (i.e. low V_{DS}):

$$\frac{I_{ON}}{W} = \frac{1}{L} \mu_{gm} C_{BG} (V_{BG} - V_{TH}) V_{DS} \quad [1a]$$

$$\frac{(V_{BG} - V_{TH})_1}{(V_{BG} - V_{TH})_2} = \frac{W_2}{W_1} \frac{L_1}{L_2} \frac{\mu_{gm_2}}{\mu_{gm_1}} \frac{C_{BG2}}{C_{BG1}} \quad [1b]$$

Note that $V_{DS} = 1$ V was used in both the studies. Although the expected ratio is ~ 0.95 from Eq. 1, we derived this ratio to be ~ 1.06 for $I_{ON} = 1$ $\mu\text{A}/\mu\text{m}$ from the experimental data (Fig. R1a-b). This proves the fact that our device characteristics are consistent with the previous work.

We have revised the Fig. 2c to normalize the device current to the channel width.

3. The authors claim their top-gate and back-gate can be interchanged. However, in Extended data 1d, the on/off ratio using top-gate is much lower than that using back-gate. The authors need to explain this. In addition, the reason for choosing top-gate as the presynaptic terminal (VPSV) needs to be explained. What are the disadvantages for using back-gate as VPSV?

The reviewer's observation is accurate, the on-off current ratio can be orders of magnitude lower for the top-gate voltage sweep (V_{TG}) compared to the back-gate voltage sweep (V_{BG}). In fact, the on-off ratio for V_{TG} sweep depends on V_{BG} (Fig R2a). This is because of the fact that in a top-gated geometry, MoS₂ channel underneath the contacts cannot be gated by the top-gate, whereas the global back-gate can control the entire MoS₂ channel including the extension regions underneath the contacts (Fig R2b) [11, 12]. As such, the top-gate characteristics can be severely limited by the channel resistance underneath the contacts. For instance, as the V_{BG} becomes more negative both the channel and the channel underneath the contacts are biased in the subthreshold regime, where the resistance of MoS₂ increases exponentially. While the channel can be switched to the on-state by applying positive V_{TG} values, the extensions underneath the contacts remain unaltered adding extra resistance that dominates the overall current transport. Since this extension resistance due to the MoS₂ channel underneath the contacts depend on V_{BG} so is the on-off ratio obtained during the V_{TG} sweep. Note that this phenomenon is a direct consequence of the fact that unlike Si, MoS₂ FETs lack degenerate doping underneath the contacts. Several doping strategies are being developed to circumvent these challenges, which are beyond the scope of discussion in the context of the present manuscript [13].

We have included the above discussion as *Supplementary Note 1*.

Figure R2. a) Top-gate transfer characteristics of multilayer MoS₂ FET with HSQ as the top-gate dielectric for different back-gate voltages with 285 nm SiO₂ as the back gate dielectric. b) Schematic and energy band diagram showing that the MoS₂ channel underneath the contacts cannot be gated by the top-gate, whereas the global back-gate can control the entire MoS₂ channel including the extension regions underneath the contacts.

We also thank the reviewer for asking us to highlight the advantages of using top/bottom gates as presynaptic/sampling terminals and *vice versa*. Both alternatives have their own merits based on the application. For example, different analog inputs can be encoded simultaneously if the top-gate is used as the presynaptic terminal in a chip containing multiple neural encoders with the global back-gate acting as the common stochastic sampling terminal. Similarly, the same analog input applied to the presynaptic back-gate terminal can be encoded into different spike trains by adjusting the transfer function of individual neural encoders through the top-gate sampling terminal.

We have included the above discussion in **Supplementary Figure 1**.

4. In Figure 2g, the choice of load resistor (R_L) and V_{PSV} range should be explained. Why the authors choose R_L to achieve range of 0V-5V for V_{PSV} instead of other ranges?

The reviewer has asked a pertinent question. Other ranges are equally acceptable. However, note that our objective is to keep the operating voltages to below 5-10 V to achieve low power operation as well as make them compatible for eventual deployment in commercial and defense related applications. Our design methodology for R_L is as follows:

1. Identify the desired V_{PSV} range from the dual-gated MoS_2 FET characteristics (0-5 V).
2. Measure the output current (I_{PD}) range for the Si photodiode (0-2 μA) corresponding to the input LED power range (0-80 mW).
3. Determine the required R_L for the transduction of obtained photocurrent range to the identified presynaptic voltage range.

5. In Figure 2f and 2g, the authors use different range of LED power. One is 0-200mW, and another is 0-80mW. The author should make the range the same. Otherwise, it is difficult to compare the IPD in these two figures and check whether they are consistent.

We agree with the reviewer's suggestion.

We have updated the figures in the revised version of the manuscript.

6. In Figure 3c, 3g and 3k, the authors need to explain how to define the spike, and what the I_{ST} used in each encoding algorithm.

We apologize for missing this information. For each encoding algorithm, we have used the same $I_{ST} = 500$ pA.

We have included this information in the revised manuscript.

7. In Figure 3c, 3g and 3k, the authors use a 10ms VTG pulse for current sampling. The authors need to explain how they read the current during this 10ms period, by taking average of current, or choosing any point in this period, or other methods?

The reviewer has brought up an important point. Note that all measurements in this work were done using Keysight B1500A parametric analyzer. According to the manual, when we set the sampling delay (for example, $T = 10$ ms) the tool determines a time width (W) for integration based on the current value i.e. lower current values need larger W and *vice versa* (Fig. R3). If $W < T$ the tool measures the current at any point within the delay period. Else if $W > T$, irrespective of the time delay, the tool measures the current every time width (W) it sets for the current value range. The 10ms delay we chose is large enough for the current value ranges we are operating to integrate. So, the tool will measure at any point within the 10ms delay.

[Redacted]

Figure R3. Pulsed measurements using Keysight B1500A parametric analyzer. According to the manual, when we set the sampling delay (for example, $T = 10$ ms) the tool determines a time width (W) for integration based on the current value i.e. lower current values need larger W and vice versa. If $W < T$ the tool measures the current at any point within the delay period. Else if $W > T$, irrespective of the time delay, the tool measures the current every time width (W) it sets for the current value range.

This information has been added to the **Method** section of the revised manuscript.

8. For Figure 3d, 3e, 3h, 3i, 3l and 3m, the authors should also show the standard deviation (error bar) of each point. For example, the authors take the mean spike interval to plot Figure 3d, but they also need to show the standard deviation (error bar) of the spike interval under each VPSV. Otherwise this trend line in Figure 3d is not very convincing.

We agree with the reviewer that the standard deviation values must be reported. We have included this information based on the results from 16 experiments in the revised manuscript (Fig. R4).

Figure R4. Statistics of a) spike-rate b) spike-count and c) spike-timing based encodings. The top panel shows results of all 16 experiments for different V_{PSV} . The middle panel shows the corresponding means and standard deviations (error bars) for different V_{PSV} . The bottom panel shows the corresponding means and standard deviations (error bars) for encoding energy expenditure per spike for different V_{PSV} .

We have included error bars for **Fig. 3d, 3e, 3h, 3i, 3l and 3m** in the revised manuscript.

9. In Figure 3d, 3e, 3h, 3j, 3l and 3m, the authors need to explain how they get the dashed lines. What are the functions they used to fit the points? In addition, the authors need to explain why Figure 3i and 3m do not follow linear relationship.

The reviewer's observations are correct. The dotted lines were guides to the eyes. We have removed these lines and instead used mean and standard deviation to plot the data.

We have removed the dotted lines from **Fig. 3d, 3e, 3h, 3i, 3l and 3m** in the revised manuscript.

10. In Figure 3n-q, the authors need to explain how the images are encoded in detail. (i) It seems that the authors only have results at VPSV range of 0 to 5V with step of 0.5V. If pixel values are linearly mapped to VPSV range of 0 to 5V, what are the spike rate, spike count and spike timing at other VPSV values? For example, VPSV = 0.02 V. (ii) What do the pixel values in Figure 3o-q represent for? If the pixel values in the encoded images refer to spike rate, spike count and spike timing, Figure 3q should be an inverse image, since larger VPSV leads to small spike time value (Figure 3l).

The reviewer has raised an excellent point. We agree that more detail must be included to explain how the cameraman image was encoded. Note that the gray scale pixel values in an 8-bit cameraman image range from 0-255, which are mapped to V_{PSV} range of 0-5 V. This would require a V_{PSV} precision of $5/255 = 0.0196 \approx 0.02$ V. However, our experimental V_{PSV} step size was 0.5 V. Therefore, we quantized the cameraman image into 11 distinct levels and correlate those levels to $V_{PSV} = 0, 0.5, 1.0, 1.5, 2.0, 2.5, 3.0, 3.5, 4.0, 4.5,$ and 5.0 V. Note that for spike-rate based encoding, our maximum frequency is 100 Hz and the average standard deviation for the encoded frequencies i.e. the encoding error bar is ~ 10 Hz (Fig R5a), allowing distinct encoding of 11 levels. One way to improve the mapping precision for spike-rate based encoding is to increase the maximum encoding frequency to 1000 Hz through faster sampling assuming that the encoding error bar remains unaltered. Similarly, for spike-count based encoding, the maximum count is 16 and the average standard deviation for the encoded count or the encoding error bar is ~ 1.5 (Fig R5b), again allowing distinct encoding of 11 levels. Finally, for spike-timing based encoding, the maximum number of time-steps is 30 and the average standard deviation for the encoded time-step or the encoding error bar is ~ 4 (Fig R5c), allowing distinct encoding of 8 levels. The mapping precision for spike-count and spike-timing based encoding can be increased by increasing the total number of sampling points.

Figure R5. Encoding precision (error bar or standard deviation) for a) spike-rate b) spike-count and c) spike-timing based encoding for different V_{PSV} .

The above discussion is included in the *Method* section in the revised manuscript.

We are sorry for not providing the information about the pixel values in Fig. 3o-q. We have added the color bars representing the spike-rate, spike-count, and spike-time in Fig. 3o-q (Fig. R6). Also, the reviewer is right that the Fig. 3q should be the inverse image since the higher pixel values should spike earlier than the lower pixel values for spike-time based encoding. However, the information we encoded in the spike-time encoding is $(T - t_{spike})$, where T is the total time-period and t_{spike} is the timing of the first spike, which will make it proportional to the input pixel value. However, we have revised Fig. 3q following the reviewer's recommendation (Fig. R6d).

Figure R6. a) Original, b) spike-rate encoded, b) spike-count encoded, and d) spike-timing encoded Cameraman image. The pixel values ranging from 0 to 255 were mapped linearly to the V_{PSV} range of 0 to 5 V. Clearly, irrespective of the encoding algorithm the Cameraman image is accurately encoded.

We have revised **Fig. 3n-q** in the revised manuscript.

11. In Extended data 6, the authors use Lena image to compare the correlation coefficient (CC), which is not consistent with the main text.

We are sorry for the confusion.

We have updated the figures in the Extended data 6 (now *Supplementary Figure 6*).

12. The authors should list a table for all the related parameters they used for the encoding transfer function (especially for Figure 4a, 4c, 4e), including VTG average, VTG standard deviation and IST. Otherwise, it is difficult to compare whether Figure 4a, 4c and 4e are consistent or not.

We agree with the reviewer's suggestion.

We have revised Fig. 4a-f to include the requested information.

13. Is there any full lobe of the transfer curve for both top-gate and back-gate? The authors need to show that there is no hysteresis in this dual gated transistor. Otherwise, hysteresis can significantly influence the sampling results depending on the previous pulse amplitude.

We agree with the reviewer's suggestion. The characterization of hysteresis in the dual-gated device is shown in Fig. R7.

Figure R7. Dual sweep a) back-gate and b) top-gate transfer characteristics of dual-gated multilayer MoS₂ FET.

We have included the characterization of hysteresis in the dual-gated device in the **Supplementary Figure 7**.

14. What is the thickness of MoS₂ used in this work? Will the thickness affect the dual gate control?[2]

The reviewer has a valid concern. The thickness of MoS₂ flake used in this study is ~5 nm. The thickness of MoS₂ can influence the dual gate control if the flake is too thick due to screening of charges [13, 14]. If the flake is too thick, the top-gate will be unable to control the layers close to the bottom-gate and *vice-versa*, creating shunting paths, which will impact the device off-state, and subthreshold characteristics. We have included the flake thickness in the revised manuscript.

References

- [1] I. Gupta, A. Serb, A. Khiat, R. Zeitler, S. Vassanelli, and T. Prodromakis, "Real-time encoding and compression of neuronal spikes by metal-oxide memristors," *Nat Commun*, vol. 7, p. 12805, Sep 26 2016.
- [2] C. Zhao, B. T. Wysocki, Y. Liu, C. D. Thiem, N. R. McDonald, and Y. Yi, "Spike-Time-Dependent Encoding for Neuromorphic Processors," *ACM Journal on Emerging Technologies in Computing Systems*, vol. 12, pp. 1-21, 2015.
- [3] I. Sourikopoulos, S. Hedayat, C. Loyez, F. Danneville, V. Hoel, E. Mercier, *et al.*, "A 4-fJ/Spike Artificial Neuron in 65 nm CMOS Technology," *Front Neurosci*, vol. 11, p. 123, 2017.
- [4] A. Sengupta, P. Panda, P. Wijesinghe, Y. Kim, and K. Roy, "Magnetic Tunnel Junction Mimics Stochastic Cortical Spiking Neurons," *Sci Rep*, vol. 6, p. 30039, Jul 21 2016.
- [5] S. Ambrogio, N. Ciocchini, M. Laudato, V. Milo, A. Pirovano, P. Fantini, *et al.*, "Unsupervised Learning by Spike Timing Dependent Plasticity in Phase Change Memory (PCM) Synapses," *Front Neurosci*, vol. 10, p. 56, 2016.
- [6] A. J. Hill, J. W. Donaldson, F. H. Rothganger, C. M. Vineyard, D. R. Follett, P. L. Follett, *et al.*, "A Spike-Timing Neuromorphic Architecture," presented at the 2017 IEEE International Conference on Rebooting Computing (ICRC), 2017.
- [7] M. Suri, D. Querlioz, O. Bichler, G. Palma, E. Vianello, D. Vuillaume, *et al.*, "Bio-Inspired Stochastic Computing Using Binary CBRAM Synapses," *IEEE Transactions on Electron Devices*, vol. 60, pp. 2402-2409, 2013.
- [8] D. S. Schulman, A. J. Arnold, and S. Das, "Contact engineering for 2D materials and devices," *Chem Soc Rev*, Mar 2 2018.
- [9] S. Das, H. Y. Chen, A. V. Penumatcha, and J. Appenzeller, "High performance multilayer MoS2 transistors with scandium contacts," *Nano Lett*, vol. 13, pp. 100-5, Jan 09 2013.
- [10] C. D. English, G. Shine, V. E. Dorgan, K. C. Saraswat, and E. Pop, "Improved Contacts to MoS2 Transistors by Ultra-High Vacuum Metal Deposition," *Nano Lett*, vol. 16, pp. 3824-30, Jun 8 2016.
- [11] J. R. Nasr and S. Das, "Seamless Fabrication and Threshold Engineering in Monolayer MoS2 Dual-Gated Transistors via Hydrogen Silsesquioxane," *Advanced Electronic Materials*, vol. 5, p. 1800888, 2019.
- [12] J. R. Nasr, D. S. Schulman, A. Sebastian, M. W. Horn, and S. Das, "Mobility Deception in Nanoscale Transistors: An Untold Contact Story," *Advanced Materials*, vol. 31, p. 1806020, 2019.
- [13] A. J. Arnold, D. S. Schulman, and S. Das, "Thickness Trends of Electron and Hole Conduction and Contact Carrier Injection in Surface Charge Transfer Doped 2D Field Effect Transistors," *ACS Nano*, vol. 14, pp. 13557-13568, 2020/10/27 2020.
- [14] D. Saptarshi and A. Joerg, "Screening and interlayer coupling in multilayer MoS2," *physica status solidi (RRL) – Rapid Research Letters*, vol. 7, pp. 268-273, 2013.

REVIEWERS' COMMENTS

Reviewer #1 (Remarks to the Author):

The authors have answered all of my concerns and made a comprehensive revision.

Reviewer #2 (Remarks to the Author):

The authors have carefully addressed all concerns, and the revised manuscript is strengthened substantially. On such basis, I would like to recommend the acceptance of this manuscript in its present form in Nature Communications.